# Tactile Sensibility Thresholds in Implant Prosthesis, Complete Dentures and Natural Dentition: Review about Their Value in Literature

**DOI:** 10.3390/medicina58040501

**Published:** 2022-03-31

**Authors:** Diego González-Gil, Javier Flores-Fraile, Joaquín López-Marcos

**Affiliations:** Dental Clinic Faculty of Medicine, University of Salamanca, Paseo Universidad de Coimbra SN, 37007 Salamanca, Spain; diegoggil@usal.es (D.G.-G.); jflmarcos@usal.es (J.L.-M.)

**Keywords:** osseoperception, tactile sensibility, interocclusal thickness, interocclusal perception

## Abstract

*Background and Objectives:* The periodontium has important proprioceptive receptors that prevent teeth from using excessive occlusal forces during chewing. There are other receptors from adjacent tissues that replace periodontal ones when teeth are extracted and rehabilitated with prosthesis, although they seem to be less effective. Psychophysical studies investigate tactile sensibility thresholds, which are useful to measure this masticatory efficiency in different prosthetic rehabilitations. There are two types of sensibility depending on the receptors that are activated during these studies: active and passive tactile sensibility. The purpose of this study is to obtain active and passive tactile sensibility threshold figures in natural dentition and prosthetic rehabilitations so we can compare them and understand how this sensibility works in different situations. *Materials and Methods:* We performed a systematic review of the available literature, following PRISMA guidelines and including articles from 2004 to 2021 in the MEDLINE database. Only 10 articles were included in this investigation as they provided concrete threshold figures. *Results:* The mean values of active tactile sensibility thresholds in complete dentures, implant prosthesis and natural dentition are 64 µ, 23.3 µ and 16.1 µ, respectively. The mean values of passive tactile sensibility thresholds in implant prosthesis and natural dentition are 6.7 N and 0.8 N, respectively. *Conclusions:* Implant prosthesis have lower thresholds, that are very close to those present in natural dentition, than complete dentures due to an increased tactile sensibility. Active tactile sensibility thresholds present fewer differences between values than passive tactile ones; as these are only influenced by receptors from periodontal or periimplant tissues.

## 1. Introduction

The periodontal ligament (PDL) is a great source of sensitive information inside the oral cavity as it contains several kinds of mechanoreceptor. These receptors are very important in the functioning of the masticatoy system. One of their main functions is avoiding excessive forces during occlusal chewing so as not to harm teeth. They notice small pressure or position changes that may happen in the interocclussal space, or even the toughness of food inside our mouth; then, this information is quickly transmitted to the brain [1,2,3].

Edentulous patients have lost periodontal receptors and the important proprioceptive information associated with them. When these patients are rehabilitated with complete dentures, other receptors from adjacent tissues, such as muscles or temporomandibular joints, are stimulated during chewing. This provides useful proprioceptive information which is known as tactile sensibility [3,4,5,6,7,8]. This tactile sensibility appears to be increased in patients wearing implant prosthesis due to a sensitive phenomenon called osseoperception [9,10,11,12,13]. Osseoperception is the sensation coming from the mechanical stimulation of implant-supported prosthesis when receptors from peri-implant bone are activated, as well as other receptors from surrounding tissues [14,15,16,17]. A Corpas investigation [18] demonstrated the presence of nerve fibers surrounding osseointegrated implants that might be responsible for this increased tactile sensibility. Neurophysiological studies have tried to demonstrate how the brain changes when we rehabilitate a tooth with an implant, in order to understand the complex phenomenon of osseoperception [19,20,21,22]. For example, Habre-Hallage [23] performed a relevant investigation with functional magnetic resonance imaging (fRMI) that showed how there is a response in the cortical brain after the mechanical stimulation of implant prosthesis. This study also suggests that after tooth extraction and its rehabilitation with implants, brain plasticity seems to occur. Although these investigations are helpful to understand osseoperception, this phenomenon is not well known yet.

Tactile sensibility allows us to evaluate the efficiency of the masticatory system in different occlusal settings [24,25,26]. This sensibility may be passive or active. While passive tactile sensibility only measures receptors from periodontium or peri-implant tissues, active tactile sensibility also evaluates remote receptors, such as those that come from temporomandibular joint or masticatory muscles. Besides this, passive tactile sensibility is measured by Newtons, representing the minimal strength that an implant or a tooth can perceive. Active tactile sensibility is measured by micrometers (µ) as it evaluates the perception of interoclussal thickness that can be noticed by a patient during chewing [14,27].

Psychophysical investigations study both active and passive tactile sensibility in order to obtain their thresholds and to know how different receptors are stimulated. Active tactile sensibility pshichophysical studies are more common, as their functioning is easy to perform on a dental chair. Thin foils with different thicknesses are placed interocclusally, so patients can state if they are able to notice them during chewing or not. Passive–active tactile sensibility studies are more complex, as they are based on the mechanical stimulation of implants or teeth by special devices designed to apply forces directly to them. These devices punctuate implants and teeth at different intensities, then patients have to affirm whether they can perceive this mechanical stimulation [27,28].

In this context, the purpose of this study was to obtain all of the tactile sensibility threshold figures available in the literature. Besides calculating the mean values of these figures and comparing them, the aim was to understand the functioning of tactile sensibility in natural dentition and in other prosthetic situations.

## 2. Materials and Methods

### 2.1. Study Design

We have carried out a review of all psychophysical studies that showed concrete figures of tactile sensibility thresholds, including articles from the period from January 2004 to January 2021 covering only articles published in English.

We performed a study selection according to the PRISMA (Preferred Reporting Items for Systematic Review and Meta-Analyses) guidelines for reporting systematic reviews. CRD42022302552 Prospero Register Code

The search strategy was conducted using the population, intervention, comparison, and outcome (PICO) framework based on the following question: “What are the mean tactile sensibility thresholds in natural dentition, implant prosthesis and complete dentures?”

### 2.2. Inclusion Criteria

We have only included articles specifying active and passive tactile sensibility thresholds, as well as articles written in English and with the full text available.

### 2.3. Exclusion Criteria

Studies that did not show figures of tactile sensibility thresholds were rejected as the main objective of this investigation was a comparison of these values. This process is shown in the flowchart (Figure 1)

### 2.4. Variables

After obtaining the tactile sensibility thresholds, we established a comparison between them in order to know the differences in every prosthetic and natural situation studied.

### 2.5. Resources

#### Bibliographical Resources

Medical Database Pubmed-Medline was consulted, and social media ResearchGate was used as a complement in order to obtain some full-text articles under consent from their authors.

The key words used were “osseoperception”, “tactile sensibility, ‘‘interocclusal thickness’’ and ‘‘interocclusal perception’’. A cross-search was also performed using these terms: “osseoperception and tactile sensibility”.

## 3. Results

After the systematic review, only 10 articles published from 2004 to 2019 provided real figures of passive and active tactile sensibility. All of them are pshychophysical studies in humans that show the thresholds in Newtons or micrometers depending on the type of sensibility. They also study the differences of these thresholds between different kinds of prosthesis and natural dentition. Even though the methodology of these investigations is not homogeneous, we have calculated the mean values of each one in order to obtain a useful reference for every threshold. This will help us to better find out the functioning of tactile sensibility in different situations and to have a starting point to develop new studies. The mean values of tactile sensibility thresholds are shown in the tables.

Table 1 shows the main characteristics of every investigation in order to obtain results that we can measure hereafter. It shows what kind of tactile sensibility was studied and what kind of prosthetic situation was involved.

Table 2 refers to clinical investigations that studied passive tactile sensibility (PTS) thresholds in natural dentition and implant prosthesis, measuring the differences in values between them.

Table 3 measures the values in active tactile sensibility thresholds (ATS) in implant prosthesis and natural dentition.

Table 4 shows the mean values of active tactile sensibility (ATS) thresholds in complete dentures.

Finally, Table 5 provides the total mean values of ATS in every prosthetic rehabilitation studied, as well as natural dentition.

## 4. Discussion

Psychophysical studies concerning passive tactile sensibility are very scarce as they are more difficult to perform. There are only two studies [28,36] that measure this figure as it is necessary to design a complex device that applies pressure to dental implants and teeth. This pressure is measured by Newtons and is controlled by a computer. After the activation of these devices, receptors from periodontal or peri-implant tissues are stimulated, and, depending on the pressure level applied, the patient will either be able to notice a stimulus or not.

On the other hand, psychophysical studies measuring active tactile sensibility are easier to reproduce on the dental chair and are more common in the literature. These studies are based on the minimum interocclusal thickness that can be noticed by a patient with natural dentition or rehabilitated with some kind of prosthesis [30]. The methodology of these investigations is based on placing metal foils with different thicknesses interocclusally. These foils are very thin and are measured in µ. Next, the patient bit these foils so we could find out if they were able to perceive them between their teeth. A great limitation in this study is the big variation of data about similar measurements. Every investigation has its own protocol, there are different scientific samples and the statistical analyses varies. Moreover, metal foils are made of different materials and thicknesses change in every study. This explains why the active tactile sensibility thresholds are so different in each case.

In this case, not only receptors from periodontal or peri-implant tissues are activated but also other receptors from adjacent tissues, and temporomandibular joints or masticatory muscles are stimulated too [27,28,29,30,31,32,33,34,35,36].

The clinical implications of this investigation are very relevant as implant rehabilitation is a very common treatment nowadays. These results reflect how prosthesis can influence masticatory function. Beside this, it can be demonstrated how complete dentures are less effective than implant prosthesis at perceiving small occlusal forces during chewing.

After this review, we have verified that both active and passive tactile sensibility thresholds in implants are slightly reduced with respect to those present in natural dentition. Beside this, the difference between values in passive tactile sensibility is increased. This is because proprioceptive sensibility is only activated by mechanoreceptors from peri-implant tissues, without taking into account the other receptors from joints or muscles [37,38,39,40]. However, we can state that tactile sensibility thresholds in implants have values that are very close to those present in natural dentition. Thus, implant prosthesis can achieve a great adaptation and operation.

It has been a great challenge to find tactile sensibility thresholds in complete dentures as there are very few psychophysical studies on this subject, and few investigations comparing these values with tactile sensibility thresholds in implants or natural dentition. Besides this, there are no values of passive tactile sensibility in complete dentures, as these prosthesis do not have periodontal or peri-implant receptors associated with them that can be stimulated during psychophysical studies. Therefore, psychophysical studies in complete dentures must focus on receptors from surrounding tissues such as muscles, mucous membranes and temporomandibular joints. The stimulation of these receptors is measured by active tactile sensibility [3,14].

Grieznis’ investigation [28] set the active tactile sensibility thresholds in implants and natural dentition at 2.5 N and 0.8 N, respectively. The difference between these values is 1.7 N. There is a previous study [36] that also measures the active tactile sensibility threshold in implants (10.9 N) without studying this same value in natural dentition. The difference between these values is so big because the methodology of both studies is not homogeneous. Concretely, the scientific sample is very different. Every investigator uses different devices and the prosthesis measured are not the same. Therefore, after calculating the mean values of thresholds in implant prosthesis the resulting figure is 6.7 N.

We have only found two investigations comparing active tactile sensibility thresholds between complete dentures, implant prosthesis and natural dentition [30,34]. Batista said that active tactile sensibility threshold in complete dentures was 92 µ, while Shala set this figure at 36 µ. The mean value between them is 64 µ. Although we know there is limited data, there is no other way to calculate this threshold in order to improve this lack of content in future investigations. However, there is an approximate figure that helps us to set up the thickness of the interocclusal metal foils of these studies.

According to these figures, the difference between thresholds in complete dentures and implant prosthesis is 40.6 µ. This value is almost six times higher than the difference between thresholds in implant prosthesis and natural dentition (7.2 µ), which highlights the fact that there is an increased tactile sensibility in implant prosthesis with respect to complete dentures.

There are more studies investigating active tactile sensibility in implants and natural dentition [27,29,30,31,32,33,34,35]. After studying the thresholds in every article, the mean value of active tactile sensibility threshold in implants is 23.3 µ, and in natural dentition it is 16.1 µ. Thus, the difference between both values is only 7.1 µ. Those thresholds are closer to those from complete dentures and implant prosthesis.

Tactile sensibility is an important parameter during the evaluation of prosthetic rehabilitation’s efficiency. This proprioceptive information protects us from applying an excessive chewing force when we are eating. Without this sensibility we could harm our teeth or break our prosthesis while biting hard food. Although the differences in values of active tactile sensibility between implants and natural dentition is small, it is enough to induce fractures in prosthesis because of this deteriorated masticatory function. As the tactile sensibility in implants is increased and its thresholds are similar to those present in natural dentition, implant-supported prostheses are a great alternative to complete dentures, which have a higher threshold and for which the proprioceptive information is limited.

## 5. Conclusions

According to the available literature, passive tactile sensibility thresholds are 6.7 N in implant prosthesis, and 0.8 N in natural dentition. After calculating the mean values of active tactile sensibility thresholds, these figures are: 64 µ in complete dentures, 23.3 µ in implant prosthesis, and 16.1 µ in natural dentition.

Implant prosthesis have lower thresholds, which are very close to those present in natural dentition compared to complete dentures due to an increased tactile sensibility related to osseoperception. As active tactile sensibility stimulates a higher number of receptors compared with passive tactile sensibility, its values present fewer differences between thresholds than passive tactile ones, as these are only influenced by receptors from periodontal or peri-implant tissues. These figures can be useful as a starting point in order to develop more studies with a homogeneous methodology that may help to understand tactile sensibility in a clearer way.

## Figures and Tables

**Figure 1 medicina-58-00501-f001:**
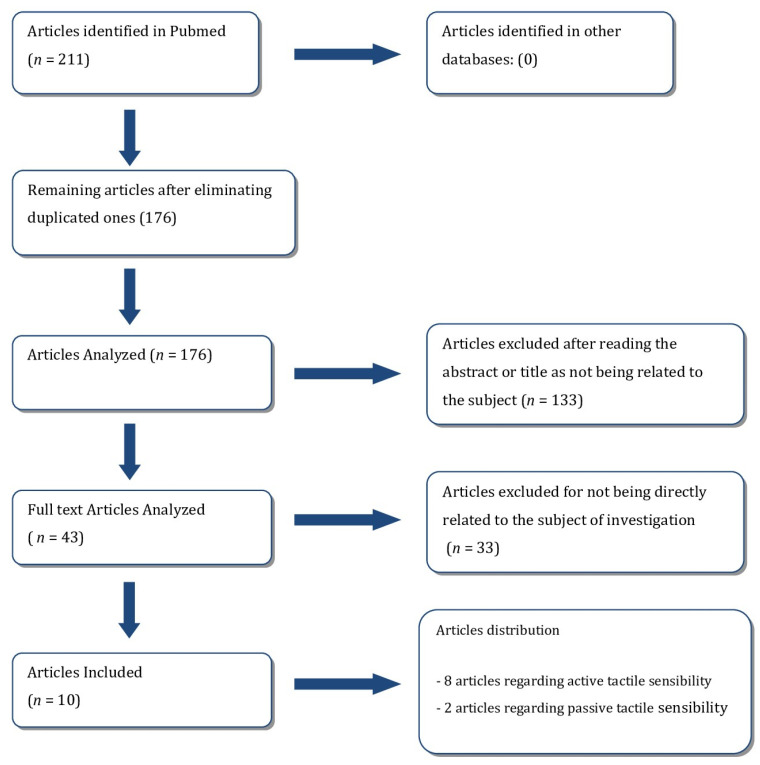
Pubmed Flowchart summarizing the review process.

**Table 1 medicina-58-00501-t001:** Showing which kind of sensibility and prosthesis was involved in each investigation.

Author and Year	Tactile Sensibility Studied	Natural Dentition Thresholds	Implant Prosthesis Thresholds	Complete Dentures Thresholds
Negahdari, 2019 [29]	Active	Yes	Yes	No
Shala, 2017 [30]	Active	Yes	Yes	Yes
Kazemi, 2013 [31]	Active	Yes	Yes	No
Reveredo, 2013 [32]	Active	Yes	Yes	No
Enkling, 2012 [33]	Active	Yes	Yes	No
Grieznis, 2010 [28]	Passive	Yes	Yes	No
Enkling, 2010 [27]	Active	Yes	No	No
Batista, 2008 [34]	Active	Yes	Yes	Yes
Enkling, 2007 [35]	Active	Yes	Yes	No
El Sheik, 2004 [36]	Passive	No	Yes	No

**Table 2 medicina-58-00501-t002:** Passive tactile sensibility (PTS) thresholds in natural dentition and implant prosthesis.

Author and Year	Mean Value of PTS Threshold in Implants	Mean Value of PTS Threshold in Natural Dentition	Difference between Values
El Sheikh, 2004 [36]	10.9 N	-	-
Grieznis, 2010 [28]	2.5 N	0.8 N	1.7 N

**Table 3 medicina-58-00501-t003:** ATS in implant prosthesis and natural dentition.

Author and Year	Mean Value of ATS Threshold in Implants	Mean Value of ATS Threshold in Natural Dentition
Negahdari, 2019 [29]	33.1 µ	24.9 µ
Shala, 2017 [30]	30.5 µ	13.5 µ
Kazemi, 2013 [31]	30 µ	21.4 µ
Reveredo, 2013 [32]	24 µ	12 µ
Enkling, 2012 [33]	20 µ	16.9 µ
Enkling, 2010 [27]	20.2 µ	-
Batista, 2008 [34]	12 µ	10 µ
Enkling, 2007 [35]	16.7 µ	14.3 µ
**Total Mean Value of ATS Threshold in Implants**	**Total Mean Value of ATS Threshold in Natural Dentition**	**Difference between Values**
23.3 µ	16.1 µ	7.2 µ

**Table 4 medicina-58-00501-t004:** ATS thresholds in complete dentures.

Author and Year	Mean Value of ATS Threshold in Complete Dentures
Shala, 2017 [30]	36 µ
Batista, 2008 [34]	92 µ
Total mean value of ATS threshold in complete dentures	64 µ

**Table 5 medicina-58-00501-t005:** Total means values of active tactile sensibility (ATS).

Prosthetic Situation	Total Mean Value of ATS Threshold	Standard Deviation
Complete dentures	64 µ	39.59 µ
Implant prosthesis	23.3 µ	7.42 µ
Natural dentition	16.1 µ	5.33 µ

## Data Availability

Not applicable.

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
