# Peer review of "Tactile Sensibility Thresholds in Implant Prosthesis, Complete Dentures and Natural Dentition: Review about Their Value in Literature"

_medicina, 2022, doi:10.3390/medicina58040501_

Round 1

Reviewer 1 Report

I would like to thank the authors for the manuscript. It is very well structured, the content is short and clear, and it has a clear focus.

The introduction sounds fantastic.

The material and methods section allows the reproduction of the study, so it seems very good to me.

The results are precise, although more information could be given. The mean obtained with the values ​​of the different articles is calculated very well. Still, I find it essential to add some measurement of data dispersion, such as the standard deviation. There is a lot of variation in the values ​​obtained in the different articles, and I think that should be emphasized somehow.

The discussion seems fine to me, although it could be completed.

The significant variability of the results obtained in the different articles when looking at the same concept should be discussed more. The authors already say that the methodologies are very different, but these differences could be discussed further.

The threshold found for implant-supported prostheses and natural teeth appears to be very similar. It would be interesting for the authors to discuss why there is a higher fracture incidence of the restorative material in patients rehabilitated with bimaxillary implant-supported fixed prostheses. One of the hypotheses was the lack of proprioception when losing the periodontal ligament, but this hypothesis may be ruled out in the present manuscript. It would be helpful to discuss what theories could cause this higher incidence of complications.

Thank you very much for your knowledge and your need to share it with the professional community. Congratulations on the manuscript.

Author Response

Dear reviewer, I want to thank you for your words, advices and corrections.

This manuscript has been reviewed by a native English speaker in order to solve every grammatic mistake.

Also, we have added more information about data dispersion; the standard deviation has been placed besides the medium values in their table. We have completed the missing information about the different methodologies in clinical investigations and why the values present this great variation. We have explained why implant prosthesis have a high incidence of fractures too.

Thank you for your consideration!

Reviewer 2 Report

This manuscript is titled "Tactile sensibility thresholds in implant prosthesis, complete dentures and natural dentition: review about their value in literature".

My comments are as below:

  • The manuscript has several grammatical errors and spelling mistakes throughout that needs to be reviewed and corrected by an English language expert.
  • Authors should list the study limitations in the Discussion section.
  • Authors should clearly outline the clinical implications of the study results.
  • Table titles need to be a bit more elaborate as Tables are considered as "stand alone" entities and should be able to provide adequate information to the reader if looked upon separately. In addition, please expand all abbreviations used in the Tables.

Author Response

Dear reviewer, I want to thank you for your words, advices and corrections.

This manuscript has been reviewed by a native English speaker in order to solve every grammatic mistake.

Also, we have detailed the limitations of this investigation as well as its clinical implications. We have completed the missing information about the different methodologies in clinical investigations and why the values present this great variation. We have changed the information about the tables, so they will be easier to understand too.

Thank you for your consideration!
